# Spatio-Temporal Encoding of Brain Dynamics with Surface Masked Autoencoders

**Simon Dahan**[1]                                                    SIMON.DAHAN@KCL.AC.UK
**Logan Z. J. Williams**[1,2]                                   LOGAN.WILLIAMS@KCL.AC.UK
**Yourong Guo**[1,2]                                             YOURONG.GUO@KCL.AC.UK
**Daniel Rueckert**[3]                                          DANIEL.RUECKERT@TUM.DE
**Emma C. Robinson**[1,2]                                    EMMA.ROBINSON@KCL.AC.UK

[1] *Department of Biomedical Engineering & Imaging Science, King's College London*

[2] *Centre for the Developing Brain, King's College London*

[3] *Institute for AI in Medicine, Technical University of Munich*

**Editors:** Accepted for publication at MIDL 2024

## Abstract

The development of robust and generalisable models for encoding the spatio-temporal dynamics of human brain activity is crucial for advancing neuroscientific discoveries. However, significant individual variation in the organisation of the human cerebral cortex makes it difficult to identify population-level trends in these signals. Recently, Surface Vision Transformers (SiTs) have emerged as a promising approach for modelling cortical signals, yet they face some limitations in low-data scenarios due to the lack of inductive biases in their architecture. To address these challenges, this paper proposes the surface Masked AutoEncoder (`sMAE`) and video surface Masked AutoEncoder (`vsMAE`) - for multivariate and spatio-temporal pre-training of cortical signals over regular icosahedral grids. These models are trained to reconstruct cortical feature maps from masked versions of the input by learning strong latent representations of cortical structure and function. Such representations translate into better modelling of individual phenotypes and enhanced performance in downstream tasks. The proposed approach was evaluated on cortical phenotype regression using data from the young adult Human Connectome Project (HCP) and developing HCP (dHCP). Results show that `(v)sMAE` pre-trained models improve phenotyping prediction performance on multiple tasks by $\geq 26\%$, and offer faster convergence relative to models trained from scratch. Finally, we show that pre-training vision transformers on large datasets, such as the UK Biobank (UKB), supports transfer learning to low-data regimes. Our code and pre-trained models are publicly available at https://github.com/metrics-lab/surface-masked-autoencoders

**Keywords:** Vision Transformers, Cortical Analysis, fMRI Encoding, Geometric Deep Learning

## 1. Introduction

The construction of robust and generalisable AI models of human brain function remains a formidable challenge due to the high-dimensional, temporal fluctuations of human brain activations (Vidaurre et al., 2017; Pervaiz et al., 2022). These patterns exhibit considerable heterogeneity across individuals, making it difficult to learn latent representations that generalise across individuals. Focusing specifically on the cerebral cortex, research has long

shown that different areas perform different functions (Glasser et al., 2016), and that high-order cognition arises from dynamic interactions between these regions (Owen et al., 2021). For these reasons, several studies have chosen to model the brain as a graph (Bullmore and Sporns, 2009) and study functional dynamics using graph neural networks or sequence models (Dahan et al., 2021; Choi et al., 2023; Kim et al., 2023). One limitation with this approach resides in the difficulty of delineating cortical functional areas from MRI. Most studies assign regions from a population-average atlas (Kim et al., 2021); however, this inserts noise and errors into the estimation of regional timeseries, since human brains cannot be perfectly spatially normalised to a template through diffeomorphic registration (Glasser et al., 2016). Other studies instead prefer to treat brain activity independently at each voxel (Huth et al., 2016), but this ignores the spatial coherence of signals, both from adjacent voxels that belong to the same region, as well as distantly connected areas whose time series are correlated with that region. Pioneering work by Fischl et al. (1999), but later advanced by the HCP (Glasser et al., 2013, 2016; Coalson et al., 2018), proved that analysis of cortical fMRI is most precise when treated as functions on a surface mesh. This suggests that encoding and decoding from fMRI might be improved by explicitly accounting for long-range spatial-temporal interactions across the cortical surface.

Vision transformers (Dosovitskiy et al., 2020) (ViTs) have been established as a powerful tool for studying long-range dependencies in natural images, leveraging the mechanisms of self-attention to outperform CNNs across a range of image understanding tasks (Zong et al., 2023; Liu et al., 2022). Unfortunately, such performance gains usually come at the cost of requiring very large data sets, to compensate for the relative lack of constraints on transformer architectures. To overcome this limitation, self-supervision frameworks have been developed, that pre-train networks on simpler tasks (Bao et al., 2022; Caron et al., 2021). One recent popular approach has been the development of auto-encoder frameworks which seek to reconstruct whole images from inputs that have had the majority of the image corrupted (Dosovitskiy et al., 2020) or had patches masked out (He et al., 2021).

Recently we translated the concept of ViTs to the cortical surface, by proposing a surface patching scheme derived from tessellations of regular icosahedrons (Dahan et al., 2022). Treating cortical modelling as a sequence-to-sequence learning problem was shown to outperform surface convolutional approaches on a range of phenotype prediction tasks (Zhao et al., 2019; Monti et al., 2016). In this paper, we integrate the concept of MAE self-supervision (He et al., 2021) into this Surface Vision Transformer (SiT) framework. Results show that the resulting `(v)sMAEs` learn robust and generalisable representations that significantly enhance phenotype prediction, support transfer-learning from large-open datasets (to support learning in low data regimes) and most importantly support reconstruction of cortical functional dynamics (with up to 75% missing data).

## 2. Related Works

This work extends from the Masked Autoencoder (MAE) (He et al., 2021), which learns strong visual representations through modelling reconstruction of whole images from inputs which have had the majority of their features masked out. For ViTs, images are represented from a sequence of image patches (or tokens). The success of the MAE comes from its asymmetric encoder-decoder architecture, in which the encoder processes only a fraction ($\rho$) of

the input sequence - the *unmasked* tokens - while the decoder learns to reconstruct the image at full resolution based solely on the embeddings learnt from these *unmasked* tokens. Such self-supervision facilitates the learning of robust and generalisable representations, since the complexity of the self-attention operation scales quadratically with the length of the input sequence - thus, passing fewer tokens allows for the building of much deeper encoder networks. As the objective is to use only the encoder for downstream tasks a light-weight decoder is considered sufficient for reconstruction, ensuring that the full framework remains computationally efficient. Extending this concept, VideoMAE models (Tong et al., 2022; Feichtenhofer et al., 2022) have sought to address the unique challenges of reconstructing spatio-temporal patches from successive video frames. The MAE framework has also been applied to non-Euclidean domains such as point clouds, irregular meshes and graphs (Hou et al., 2022; Liang et al., 2022). The MAE approach to self-supervision contrasts with the masked patch prediction (MPP) model proposed by (Dosovitskiy et al., 2020), which instead employs a *symmetric* encoder-decoder architecture that is trained to reconstruct the *entire* image sequence after corrupting some of the input patches through masking or swapping. This approach was adapted to the cortical surface domain in (Dahan et al., 2022). However, since its inception the MPP has been shown to be repeatedly outperformed by the MAE, which demonstrates better reconstruction, efficiency, and performance on fine-tuning tasks.

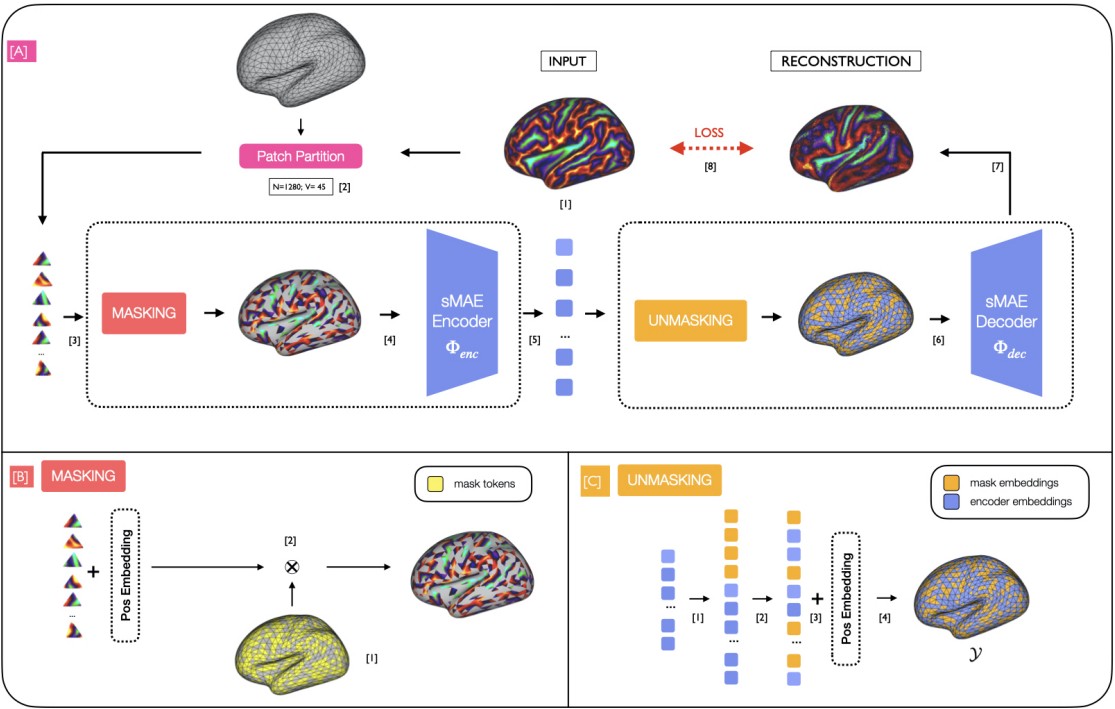

Figure 1: [A] `(v)sMAE` partitioning and learning pipelines; [B] Sequence Masking and [C] Unmasking strategies.

## 3. Methods

### 3.1. Surface Vision Transformer

Following Dahan et al. (2022), input cortical feature maps $X \in \mathbb{R}^{|V_6| \times C}$ ($C$ channels) are represented on a 6th-order icospheric (ico6) tessellation: $I_6 = (V_6, F_6)$, with $|V_6| = 40962$ vertices and $|F_6| = 81920$ faces (Figure 1.A.1). Patching is achieved by tessellating ico6 with the faces of a low-resolution icosphere, typically ico3 ($I_3 = (V_3, F_3), |F_3| = 1280$ and $|V_3| = 642$, Figure 1.A.2). This partition leads to a sequence of non-overlapping triangular patches: $T_3 = \{t_3^1, t_3^2, ..t_3^{|F_3|}\}$ (with $t_3^i \subset V_6, |t_3^i| = 45$ vertices in each patch). Imaging features (e.g. myelin and curvature) from ico6 vertices that fall within each patch are then concatenated across channels, flattened and projected with a trainable linear layer into a set of $D-$dimensional input tokens $\{X_i^{(0)}\}_{i=1}^N$. Sine-cosine positional embeddings, $E_{pos} = \{E_i\}_{i=1}^N$ are then added to each of the tokens to encode patch location within the sequence: $\mathcal{X}^{(0)} = \left[ X_1^{(0)} + E_1, ..., X_N^{(0)} + E_N \right]$, where each $E_i$ reflects a $D-$dimensional vector that encodes location from a unique combination of sine and cosine functions (Vaswani et al., 2017) (more details in Appendix A.4). Use of fixed positional embeddings, instead of the trainable embeddings used in (Dahan et al., 2022), was found to speed up network training. The initial sequence $\mathcal{X}^{(0)}$ is then processed by $L$ consecutive transformer encoder blocks of *Multi-Head Self-Attention* (MHSA) and *Feed Forward Network* (FFN) layers, with residual layers in-between:

$$\mathcal{Z}^{(l)} = \boldsymbol{MSHA}(\mathcal{X}^{(l)}) + \mathcal{X}^{(l)}$$
$$\mathcal{X}^{(l+1)} = \boldsymbol{FFN}(\mathcal{Z}^{(l)}) + \mathcal{Z}^{(l)}$$
$$= \left[ X_1^{(l+1)}, ..., X_N^{(l+1)} \right] \in \mathbb{R}^{N \times D} \tag{1}$$

Note, that sequence shape is preserved through each block. This Surface Vision Transformer (SiT) architecture forms the backbone for all (v)sMAE encoders and decoders. More information on surface patching and architecture details can be found in Appendix A.2.

### 3.2. Surface Masked AutoEncoder

Implementation of the proposed sMAE parallels that of the original MAE architecture. First, *unmasked* tokens are randomly selected from the set of all possible patches available from an ico3 mesh (Fig 1.B.1 and 1.B.2), according to the masking ratio $\rho$. These are then passed to an SiT encoder ($\Phi_{enc}$) (Fig 1.A.4), constructed from an SiT-tiny with $L = 12$ transformer blocks and 3 attention heads per layer. Next, the latent (encoder) embeddings learnt from the encoder are concatenated with a set of random (mask) embeddings - in place of the original masked tokens - to return the sequence to its original resolution $N$ (Fig 1.C.1). These are then unshuffled to restore the initial order of the sequence (Fig 1.C.2), positional embeddings are added to encode spatial information (Fig 1.C.3) and the resultant sequence $\mathcal{Y} \in \mathbb{R}^{N \times D}$ is fed to an SiT decoder ($\Phi_{dec}$) with $L = 3$ transformer blocks and 3 attention heads per layer. The last layer performs a linear projection to restore the input patch resolution ($C \times |t_3^i|$) from the sequence resolution $D$ (Fig 1.A.7). Following He et al. (2021), the network is optimised by calculating the mean square error (MSE) between the masked input feature patches and their reconstructed versions only (Fig 1.A.8).

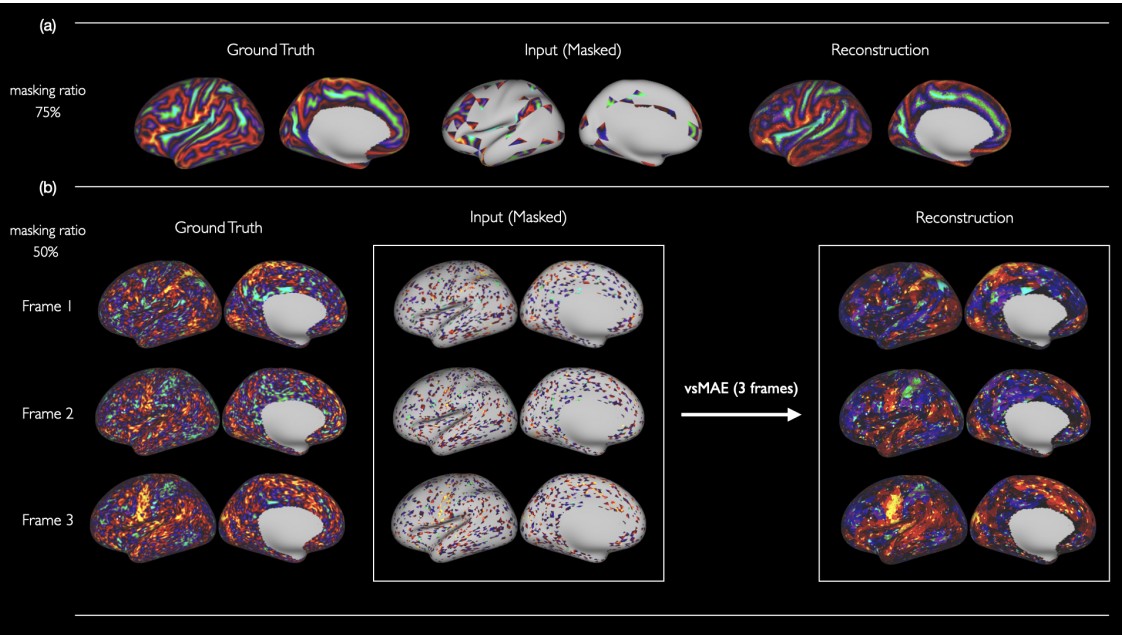

Figure 2: (a) `sMAE` sulcal depth reconstruction results on a UKB test subject ($\rho = 75\%$). (b) `vsMAE` reconstruction ($\rho = 50\%$) results with 3 7T HCP consecutive frames.

### 3.3. Video Surface Masked AutoEncoder

In the domain of video understanding, a range of spatiotemporal masking strategies have been explored for self-supervision with videoMAE. These include spacetime agnostic masking strategies (Feichtenhofer et al., 2022), which randomly sample patches in a different way for each and every frame; as well as tube-masking strategies (Tong et al., 2022; Wang et al., 2023), which sample concurrent frames from the same mask. When translating such concepts to the modelling of cortical functional dynamics it is important to acknowledge that, relative to natural video, fMRI is much less structured, with far less temporal redundancy. This is because fMRI is an indirect measure of cortical activity, characterised by low temporal resolution and corruption from physiological noise.

Thus, while agnostic masking with high masking ratios (of up to 90%) (Feichtenhofer et al., 2022) may work well for natural scenes, it cannot be easily made to work for fMRI due to the noise. Nor can classic videoMAE implementations of tube-masking, since these compress video cube patches of size $T \times 16 \times 16$ with a single linear projection. This makes sense for natural videos, for which successive frames from the same patch will probably contain highly correlated features, but not for fMRI where two frames may present very distinct patterns of activity (due to the low temporal resolution, see Figure 2.b). We address these challenges in two ways: first, we account for noise through the use of a tube-masking strategy with a reduced masking ratio (typically 50%, see Fig 2b); second, we change the approach for spatio-temporal patch compression from one projection across block of frames, to one projection per frame, followed by concatenation across frames. This design creates

a spatiotemporal sequence of tokens and enables the `vsMAE` model to effectively compute spacetime self-attention, enhancing the model's capacity to capture and integrate complex spatio-temporal dynamics between patches distant in both space and time.

## 4. Experimental Methods

To evaluate whether the proposed `(v)sMAE` can learn sufficiently rich encodings of cortical functional dynamics a series of experiments was conducted: first, we validated whether the model could robustly generalise to unseen subjects by assessing the quality of the generated reconstructions; then, we investigated whether the encodings learned by our `(v)sMAE` models capture information relevant to phenotypic predictions and therefore may suggest an alignment with neurobiological patterns. In parallel, we demonstrate that self-supervised pre-training of `(v)sMAE`s on very large open-datasets, such as UK Biobank, can support transfer-learning to data sets of much more limited size.

### 4.1. Datasets

**dHCP**   Data consists of cortical surface meshes and metrics (sulcal depth, curvature, cortical thickness and T1w/T2w myelination) derived from T1- and T2-weighted magnetic resonance images (MRI) from the developing Human Connectome Project (dHCP) (Makropoulos et al., 2018). We use 580 scans from 419 term neonates (born after 37 weeks gestation) and 111 preterm neonates (born prior to 37 weeks gestation). Preterm babies were scanned either shortly after birth, at term equivalent age (TEA), or at both timepoints.

**UKB**   Matched cortical metrics were derived from 4063 subjects (1896 biological females) aged between 46 and 83 years in the UKB Biobank (UKB) dataset (Miller et al., 2016; Alfaro-Almagro et al., 2018). These were used for the transfer learning experiment.

**HCP**   fMRI data was obtained from the movie-watching experiment of the HCP 7T release (Van Essen et al., 2013), from 174 participants who were scanned while watching a series of movie clips. fMRI responses were projected from the volume to the cortical surface (Glasser et al., 2013) and were aligned using multimodal cortical features (MSMAll) (Robinson et al., 2014a, 2018a; Glasser et al., 2016). For these subjects, we also used Z-statistic contrast maps derived from the N-back Working Memory (WM) task (Braver et al., 1997; Barch et al., 2013a) from the 3T HCP release (Van Essen et al., 2013). More details about data acquisition, processing and train/validation/test splits for all the datasets in Appendix A.1.

### 4.2. Implementation & Tasks

`(v)sMAE` **pretraining**   All models were trained on a single RTX 3090 NVIDIA GPU with Adam optimisation ($LR = 3e^{-4}$ and cosine decay). A batch size of 16 was used by default but adapted depending on $\rho$ values. The impact of different masking ratios ($\rho$) was evaluated by testing the quality of reconstructions (on all cortical features for `sMAE` and on 7T movie-task data for `vsMAE`) and fine-tuning on phenotype regression tasks. The performance of the `vsMAE` was similarly optimised over frame sampling rates $\tau \in \{1, 3, 6, 8\}$.

| Encoder architecture | Pre-training method | Nb frames used for: Pre-training/Finetuning | Sex Classification Acc (%) ± std |
|---|---|---|---|
| | None | None / 3 | 58.1 ± 0.9 |
| | sMAE | 1 / 1 | 67.1 ± 1.4 |
| SiT-tiny | vsMAE | **3 / 3** | **75.8 ± 0.5** |
| (ico3) | vsMAE | 6 / 3 | 73.2 ± 0.7 |
| | vsMAE | 8 / 3 | 75.3 ± 0.3 |

Table 1: `vsMAE` fine-tuning results on 7T frames sex classification task at different pre-training sampling rate $\tau$. Finetuning was done with a maximum of 3 frames (hardware limitations). Balanced accuracy and std averaged over 3 training runs.

**Phenotyping Predictions**   Self-supervision with `(v)sMAEs` was validated by fine-tuning the pre-trained weights from the SiT encoders on various phenotyping prediction tasks. `sMAE` encoders, trained on multivariate dHCP cortical imaging features, were validated for regression of post-menstrual age (PMA) at scan, and gestational age (GA) at birth. The GA task aims to predict the degree of prematurity from scans acquired around TEA, while the prediction of PMA was designed as a correlate for modelling healthy cortical maturation. On fMRI data, we evaluated the performance of the `vsMAE` model on sex classification. Fine-tuning was found to perform best when using SGD optimisation (momentum=0.9, warmup scheduler) with $LR = 1e^{-4}$ for sex classification, GA and PMA.

**Transfer learning**   We evaluate the potential use of `(v)sMAE` encoders for transfer learning by first training sMAEs on multivariate (sulc, curvature, cortical thickness and myelin) from UKB; the resulting encoder was then fine-tuned for dHCP PMA prediction. The `vsMAE`, pretrained on fMRI data, was fine-tuned on fluid intelligence prediction using contrast maps extracted from the HCP 3T working memory task. We used SGD optimisation (momentum=0.9, warmup scheduler) with $LR = 1e^{-5}$ for fluid intelligence prediction.

## 5. Results & Discussion

**Evaluating reconstruction of brain dynamics**   Results for the `sMAE` showed that masking ratios of $\rho = 50\%$ and $75\%$ yielded the strongest visual reconstruction, lowest reconstruction error and highest performance on GA downstream tasks (Table 3, Appendix B.1). For `vsMAE` pre-training on fMRI data, $\rho = 50\%$ provided the best trade-off between reconstruction quality and frame sampling rate $\tau$ (Table 4, Appendix B.1).

**Fine-tuning**   Results from PMA and GA experiments (Table 5, Appendix B.2.1) showed that fine-tuning `sMAE` encoders on dHCP data consistently improves their performance relative to baselines including: training from scratch; fine-tuning from ViTs pre-trained on ImageNet classification; and fine-tuning SiTs following MPP self-supervision (Dahan et al., 2022), as well as various gDL models. Notably, performance improved by 26% relative to training from scratch, which could not be achieved even with longer training times. Table 1 reports similar findings following pre-training on fMRI data, showing that sex classification significantly improves following `sMAE` pre-training on single frames. Moreover, this result

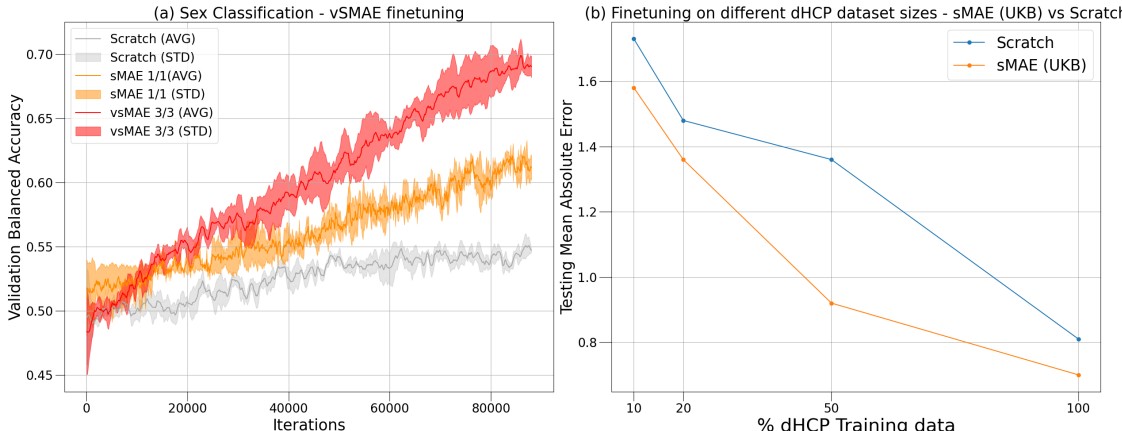

Figure 3: (a) Sex classification 7T HCP - comparing SiT models - trained from scratch (with 3 frames) (grey), fine-tuned (with 1 frame) from `sMAE` (1 frame)(orange) and fine-tuned (with 3 frames) from `vsMAE` (3 frames). (b) dHCP transfer learning experiment from `sMAE` (UKB) pre-training against SiT-tiny trained from scratch

improves by a further 13% when the spatio-temporal dynamics of the sequence is more fully taken into account by the `vsMAE` pre-training (Figure 3.a.). This suggests that modelling spatio-temporal dynamics improves cortical phenotype prediction.

**Transfer Learning** Finally, to assess the potential of `(v)sMAE`s for transfer learning on smaller datasets, we investigated fine-tuning, using subsets of the entire dHCP dataset (10%, 20% or 50%) following training on all 4063 UKB datasets. Results in Figure 3.b. show that, relative to training from scratch, transfer-learning improves cortical phenotype prediction on dHCP for all data ratios (orange line). Finetuning the `vsMAE` encoder on 3T HCP contrast maps similarly yielded higher correlation scores on the challenging fluid intelligence prediction task (0.39) Pearson correlation compared to training from scratch ($\leq 0.3$), see Appendix B.3.1.

**Discussion** In this paper, we demonstrated that pre-training surface vision transformers with `(v)sMAE`s is an effective way to learn strong representations of both static and dynamic cortical maps. Training SiTs in this way leads to better performance on downstream phenotype prediction tasks irrespective of whether the self-supervision task is trained on the same data set, or larger-open data sets such as UKB (even when the data set demographics strongly diverge). This offers significant potential for translating the benefits of SiTs to much smaller clinical neuroimaging datasets (e.g. for psychosis (Demro et al., 2021)). Moreover, the strong performance of `vsMAE` on reconstruction and phenotype prediction from fMRI suggests these models can learn robust and generalisable models of dynamic cortical function. This opens the door in future to novel applications in fMRI encoding and decoding.

## Acknowledgments

We would like to acknowledge funding from the EPSRC Centre for Doctoral Training in Smart Medical Imaging (EP/S022104/1).

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

# Appendix A. Methods

## A.1. Additional data information

### A.1.1. DHCP DATASET

We use 580 scans from 419 term neonates (born after 37 weeks gestation) and 111 preterm neonates (born prior to 37 weeks gestation). 95 preterm neonates were scanned twice, once shortly after birth, and once at term-equivalent age (TEA). For PMA prediction training data was drawn from the scans of term-born neonates and preterm neonates' first scans (26.71 to 44.71 weeks PMA). For GA, we use the scans at TEA. In both GA and PMA cases, balanced distribution of examples from each age bin was ensured (Fawaz et al., 2021).Train/validation/test splits were defined for the GA prediction task as 411/51/52, and for the PMA prediction task as 423/53/54. For the transfer learning experiment on PMA prediction, three subsets of the training data were generated with respectively 10%, 20%, and 50% of the full dHCP training dataset. The distribution of scan age across the full training set was preserved while generating the subsets. Infants were recruited and imaged for the developing Human Connectome Project (http://www.developingconnectome.org/), approved by the National Research Ethics Committee (REC: 14/LO/1169).

### A.1.2. UKB DATASET & PRE-PROCESSING

Cortical surfaces were extracted using T1w and T2w images to support accurate placement of pial surface (Glasser et al., 2013). T1w/T2w ratio maps (Glasser and Van Essen, 2011) were generated using HCP method (Glasser et al., 2013). Cortical thickness was corrected for folding-related bias as previously described (Glasser and Van Essen, 2011; Glasser et al., 2013; Sigalovsky et al., 2006). Registration of sphericalised cortical surfaces was performed using Multimodal Surface Matching (Robinson et al., 2014b, 2018b), driven by sulcal depth maps, with high regularisation (Robinson et al., 2014b, 2018b; Glasser et al., 2016). UKB was partitioned into train/validation/test splits of 2865/588/610.

### A.1.3. HCP

In the HCP 7T release (Van Essen et al., 2013), 184 participants were scanned, up to four times in separate sessions. We used data from participants who had completed all four acquisition runs (n = 174). HCP data was partitioned into train/validation/test splits of 124/25/25. On the 3T dataset, the fluid intelligence task corresponds to the number of correct responses to the Penn Progressive Matrices (PMAT) task (Barch et al., 2013b) and is known to be highly difficult to predict from medical imaging data. The same split of data was used than for the 7T HCP dataset.

## A.2. Network architecture details

In Table 2, we summarise the architecture of the (v)sMAE encoder and decoder networks used in this study and based on the SiT architecture. Here, we only used a 3-layer transformer decoder network, as it yields good reconstruction results. On a standard 24G NVIDIA GPU, the maximum batch size that can be used for vsMAE reconstruction with $\rho = 75\%$ is $\{128, 128, 64, 2\}$ for respectively $\tau \in \{1, 3, 6, 8\}$ frame-reconstruction. For the

sMAE reconstruction task, a batch size of 128 is typically used, with 4 cortical input metrics. This allows for a fast training of the `(v)sMAE` models.

| Models | Layers | Heads | Hidden size $D$ | MLP size | Params. |
|---|---|---|---|---|---|
| `(v)sMAE` encoder | 12 | 3 | 192 | 768 | 5.3M |
| `(v)sMAE` decoder | 3 | 3 | 192 | 768 | 1.4M |

Table 2: `(v)sMAE` encoder/decoder are based on the SiT architecture. All $SiT$ models preserve a hidden size of 64 per attention head. The entire encoder-decoder pipeline has only 6.7M parameters.

In the present study, we patched the cortical surface using an ico3 tessellation grid. It achieved good phenotyping performance and allow for higher-resolution patch representation, compared to ico2 sampling as in (Dahan et al., 2022), while preserving a manageable computational cost (batch size of 128 vs 256 for ico2). With the ico3 patching, the cortical surface is represented by 1280 patches of 45 vertices each, compared to 320 patches of 153 vertices for ico2.

### A.3. Masked Patch Prediction

The `sMAE` methodology is compared to the Masked Patch Prediction (MPP) self-supervision task, used previously in (Dahan et al., 2022), which in turn was adapted from (Dosovitskiy et al., 2020; Devlin et al., 2019). It employs an autoencoder architecture that is trained to reconstruct the entire image sequence while corrupting some of the input patches through masking or swapping. Following (Dosovitskiy et al., 2020), we corrupt 50% of the input patches randomly: by replacing the patches with a masked (empty) token (40%), using another patch embedding from the sequence at random (5%), or preserving their original embeddings (5%). In contrast to sMAE, the MPP methodology optimises reconstruction by computing the MSE loss for all patches, and the MPP encoder processes the entire sequence of patches, which reduces its efficiency and modelling power with long input sequences.

### A.4. Positional embeddings

Compared to (Dahan et al., 2022), we use fixed positional embeddings, which accelerate the training process compared to learned positional embeddings. Positional embeddings are defined as follows:

$$E_i = \left[ PE_{(i,j)} \right]_{j=1}^{D} \tag{2}$$

where:

$$
\begin{aligned}
PE_{(i,j)} &= sin(i/10000^{k/D}) && \text{if } j = 2k \\
PE_{(i,j)} &= cos(i/10000^{k/D}) && \text{if } j = 2k+1
\end{aligned}
\tag{3}
$$

| Masking Ratio | Reconstruction Error - sMAE | Gestational Age |
|---|---|---|
| 25% | $0.78 \pm 0.05$ | $1.51 \pm 0.1$ |
| 50% | $\mathbf{0.39} \pm 0.03$ | $\mathbf{1.35} \pm \mathbf{0.02}$ |
| 75% | $0.49 \pm 0.03$ | $1.42 \pm 0.04$ |
| 90% | $0.68 \pm 0.05$ | $1.44 \pm 0.07$ |

Table 3: Masking ratio selection. Reconstruction errors (MSE) on validation set for masked patches only. For each masking ratio configuration, the sMAE encoder was fine-tuned for GA prediction and 200 epochs. Validation prediction errors with stds across three runs are reported.

## Appendix B. Results

### B.1. Masking ratio

We first evaluate the effect of the masking ratio hyperparameter of the sMAE framework. sMAE networks were trained with different masking ratios (25%, 50%, 75% and 90%) on dHCP data and evaluated for both reconstruction quality and prediction performance in a downstream task (GA regression). Table 3 reports the best MSE reconstruction errors and mean absolute error (referred to as prediction error in the following) on the validation set, averaged across three fine-tuning runs. Figure 4 shows an example of the reconstruction quality for the validation set for each sMAE masking ratio, indicating that sMAE models capture individual cortical features even with high masking ratios, which suggests some

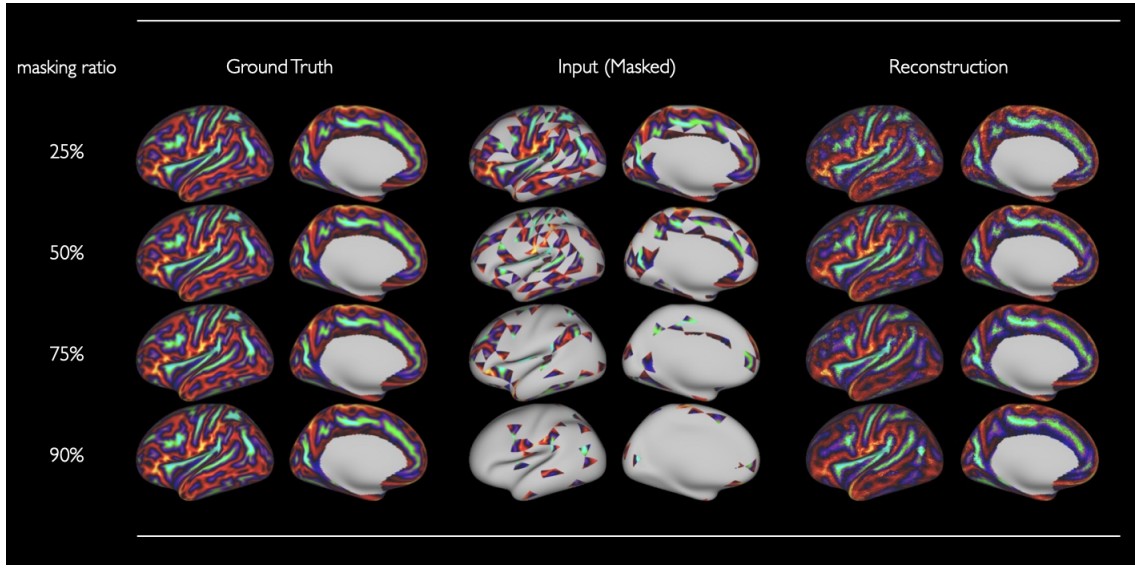

Figure 4: Sulcal depth reconstruction from sMAE pre-training at different masking ratio (25%, 50%, 75%, 90%). Results are shown for the same validation subject.

| Masking Ratio | Reconstruction Error - vsMAE (3 frames) |
|---|---|
| 50% | $\mathbf{0.55} \pm 0.03$ |
| 75% | $0.59 \pm 0.01$ |
| 90% | $0.61 \pm 0.03$ |

Table 4: Average reconstruction error on masked patches for vsMAE models pre-trained on 3-frame reconstuction.

capabilities of self-attention to model complex dependencies between brain regions. Overall, the 50% masking ratio offered the best quantitative validation and was used in all following experiments. In table 4, we report the reconstruction error for vsMAE models pre-trained on 3-frame reconstructions. Loss is averaged across masked patches only. The masking raito of 50% yield the best reconstruction error rates.

### B.2. Phenotyping prediction results

#### B.2.1. DHCP PHENOTYPING RESULTS

Results for the finetuning PMA and GA experiments of multivariate sMAE training are presented in Table 5comparing training between scratch, fine-tuning from MPP weights and fine-tuning from sMAE weights. Additionnaly, we compare the performance of the SiT models against a range of surface CNNs benchmarked in Fawaz et al. (2021). Here, we report results on the 5 most performing architectures benchmarked in Fawaz et al. (2021) on phenotyping prediction and segmentation for neonatal data: Spherical UNet (Zhao et al., 2019), MoNet (Monti et al., 2016), GConvNet (Kipf and Welling, 2017), ChebNet (Defferrard et al., 2017) and S2CNN (Cohen et al., 2018), as well as a ResNet trained on 2D projection of the spherical data. All surface CNNs were trained using the same data examples and splits as reported here. Compared to Fawaz et al. (2021), here prediction errors are averaged across 3 training runs (rather than reporting the best performance only). Finetuning the SiT encoder after sMAE pretraining outperforms all other models and training settings (Imagenet, MPP) - except for the MoNet model on PMA prediction - with smaller variations across training results (std).

---

1. (Cohen et al., 2018)
2. (Defferrard et al., 2017)
3. (Kipf and Welling, 2017)
4. (Zhao et al., 2019)
5. (Monti et al., 2016)

| Encoder Architecture | Pre-training Method | PMA at scan | GA at birth |
|---|---|---|---|
| | | error $\pm$ std | error $\pm$ std |
| Projected ResNet | ✗ | $0.97 \pm 0.34$ | $1.93 \pm 0.49$ |
| S2CNN [1] | ✗ | $0.94 \pm 0.25$ | $2.35 \pm 0.60$ |
| ChebNet [2] | ✗ | $1.21 \pm 0.49$ | $2.00 \pm 0.36$ |
| GConvNet [3] | ✗ | $0.99 \pm 0.26$ | $2.85 \pm 0.74$ |
| SUNet [4] | ✗ | $1.63 \pm 0.51$ | $2.41 \pm 0.68$ |
| MoNet [5] | ✗ | $\mathbf{0.63 \pm 0.05}$ | $1.68 \pm 0.06$ |
| SiT-tiny ico3 | ✗ | $0.87 \pm 0.08$ | $1.65 \pm 0.11$ |
| SiT-tiny ico3 | ImageNet | $0.70 \pm 0.04$ | $1.66 \pm 0.05$ |
| SiT-tiny ico3 | MPP | $0.66 \pm 0.03$ | $1.53 \pm 0.07$ |
| SiT-tiny ico3 | sMAE | $0.64 \pm 0.02$ | $\mathbf{1.22 \pm 0.04}$ |

Table 5: Fine-tuning results of PMA and GA. We compare the results with various surface CNN models and SiT training settings: from scratch, after MPP, after ImageNet or after sMAE self-supervision. Each SiT-tiny encoder is finetuned three times, averaged test prediction errors and stds are shown in the table.

### B.2.2. Sex classification Results

Sex classification training curves and loss, comparing different training regimes in the fine-tuning vsMAE experiment, are presented in Figure 5. vsMAE pre-training significantly boost the performance on sex prediction from task fMRI frames.

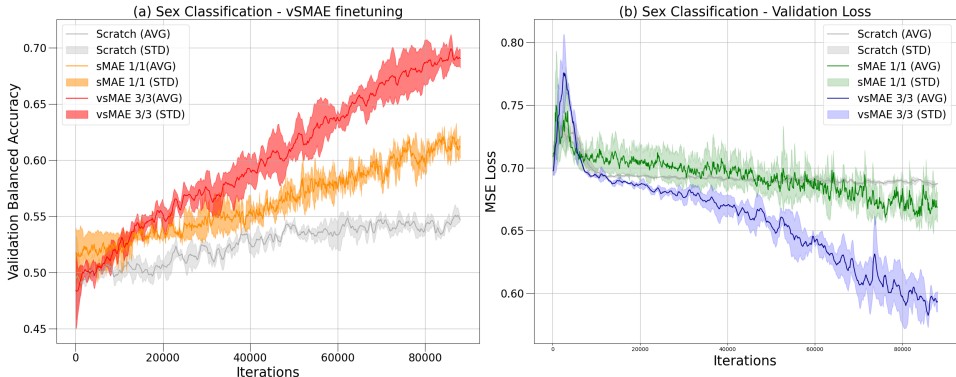

Figure 5: Validation accuracy and loss curves for the sec classification task with (v)sMAE models. (a) Three different training regimes are compared: training from scratch (with 3 frames) (gray), fine-tuned (with 1 frame) from sMAE (1 frame) (orange) and fine-tuned (with 3 frames) from vsMAE (3 frames). (b) validation loss curves for the same three training schemes. Incorporating spatio-temporal information via vsMAE pre-training and fine-tuning leads to the best results.

### B.3. Positional Embeddings

We evaluate the importance of positional embeddings, by comparing reconstructions with and without the use of positional embeddings while training the `sMAE` model. Reconstructions are presented in Figure 6. Without positional embeddings, the masked tokens can not be correctly reconstructed as no positional information of masked tokens is added to the sequence.

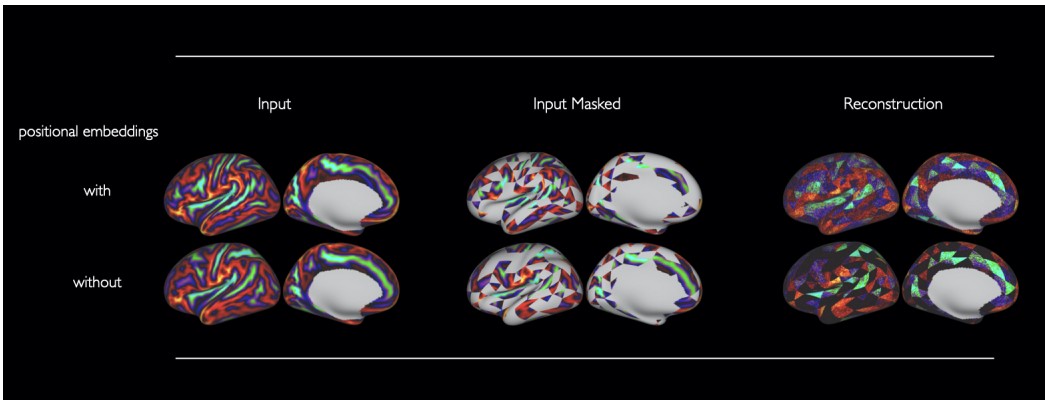

Figure 6: Reconstruction results of `sMAE` pre-training with and without the use of positional embeddings. Without positional embeddings, the model can not reconstruct the mask tokens.

### B.3.1. Fluid intelligence results

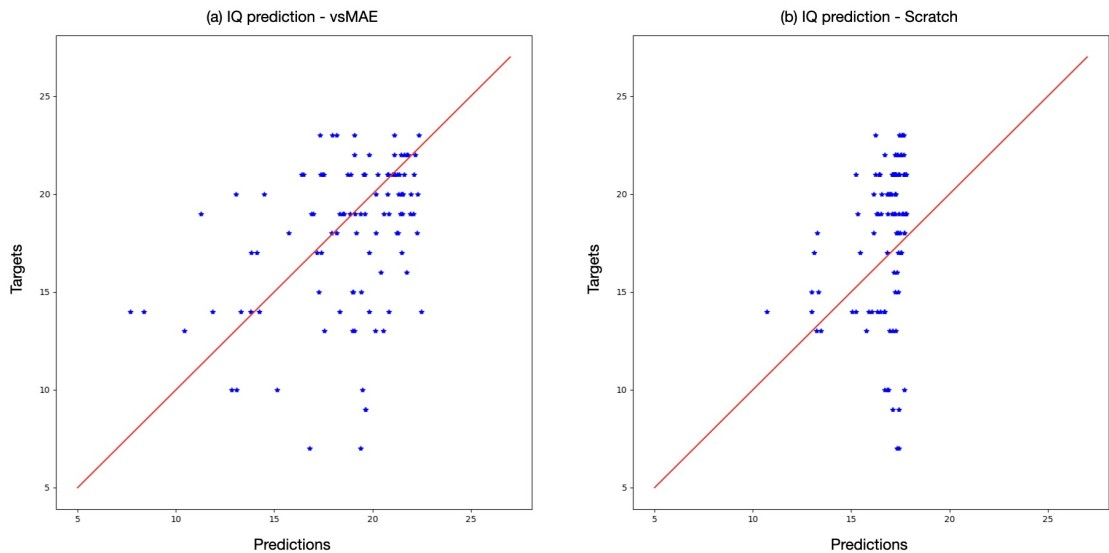

Figure 7: Fluid intelligence prediction results. (a) prediction vs targets results for a `vsMAE` model pre-trained on 7T frames and fine-tuned on 3T contrast maps. Correlation score is 0.39 (b) prediction results for a SiT-tiny trained from scratch on 3T contrast maps (correlation score of 0.3)

In Figure 7, we show the prediction results on test data of a `vsMAE` encoder fine-tuned on 3T contrasts maps for fluid intelligence, following a pre-training on 3-frame reconstruction and compared with a SiT-tiny trained from scratch on 3T contrast maps.

