# OpenReview forum: "Spatio-Temporal Encoding of Brain Dynamics with Surface Masked Autoencoders"
_MIDL.io/2024/Conference — MIDL 2024 Oral_

### Official Review · Reviewer_a431 · 2024-02-27

**Confidence:** 3
**Preliminary Rating:** 4
**Recommendation:** Oral
**Final Rating:** 5

**Summary:**

This paper proposes surface Masked AutoEncoder (sMAE) and video surface Masked AutoEncoder (vsMAE), both extensions of Surface Vision Transformers to address the data-efficiency gap during training. These models operate on multivariate and spatio-temporal cortical signals over regular icosahedral grids. These models are pre-trained to reconstruct cortical feature maps from masked inputs and learn good latent representations for downstream tasks such as prediction of cognitive measures.

They evaluate their model on cortical phenotype regression using data from the young adult Human Connectome Project (HCP) and developing HCP (dHCP) and UKBiobank for transfer learning. Their models improve phenotype prediction performance as compared to models trained from scratch and provide improvements in time to convergence.

**Strengths:**

1. The main methodology is an interesting technical extension of Surface ViT to the masked Autoencoder (MAE) and video MAE paradigm where cortical surface modeling is treated as a sequence to sequence model to generate better quality and data-efficient foundational representations for downstream tasks. The novelty of this approach is a definite plus point.

2. The authors evaluate on multiple tasks such as phenotypic prediction on multiple measures, sex classification, as well as a transfer learning task on the UKBiobank dataset

**Weaknesses:**

1. The paper reports no baseline methods apart from different version (eg. masking ratios ) of their sMAE and vsMAE apart from different sVIT models with different data used for pre-training. How do these models compare to other phenotypic prediction models used in literature? They also do not report on the data-efficiency aspect of these models as compared to sVITs, which is mentioned in the abstract as something these models address.

2. Standard deviation measures are not reported in Tables 3 and 4. Additionally, the experiments do not performance significance testing to ensure that the performance improvements are robust. Additionally, do the experiments use only one train-test-validation split or multiple?

**Detailed Comments:**

1. Something that was not clear to me from the description was the concatenation step across vertices in each patch on Page 3. Is there a certain ordering that is maintained when the representations are concatenated before feeding into the linear projection layer?

2. Have the authors quantified the reconstruction quality reported in Fig. 2 for the different masking ratios?

**Justification Of Final Rating:**

The authors have addressed a majority of my comments and improved the quality of the submission based on the suggestions provided. Overall, I think this paper is quite novel and makes a significant technical contribution to the medical imaging community and I am wiling to raise my score by a point. I hope to see it presented at MIDL

**Justification Of The Preliminary Rating:**

The methodological contribution of the paper is pretty good and the experiments demonstrate reasonable results to support the main claims of the paper. There are a few additional points mentioned under weaknesses and questions that need addressing. Therefore, my preliminary score is a weak accept.

**Questions To Address In The Rebuttal:**

Please refer to the points raised under weaknesses

**Special Issue:**

Yes

---

> ### Author Response · Authors · 2024-03-18
>
> We would like to thank the reviewer for the time they took to review our submission and their very constructive comments. We noted the important questions raised by the reviewer and tried to address them in the following.
>
> **Points to address:**
>
> *Phenotypic predictions and comparison with baseline models and other approaches*: We acknowledge the reviewer’s remarks on the lack of comparisons to baseline models for phenotyping prediction. In previous work (Fawaz et al 2022) we benchmarked the performance of surface CNNs (spectral convolution, spatial convolution) against classic ML driven by regional features derived from an atlas for the prediction of scan age (PMA) and birth age (GA) in the dHCP dataset. This showed that surface deep learning frameworks outperformed classic ML, especially for data that was not aligned (native configuration). Subsequently, in (Dahan et al 2022 MIDL 2022) we showed that the SiT improved the overall performance against these surface CNNs. In this submission, we have shown that the sMAE self-supervision of the SIT further improves performance - while, importantly, the same data splits were used throughout. We agree that a full appraisal of all these results would help readers of this paper and therefore have added these results to the appendix section in Table 5, with six baseline networks (five surface CNNs and one 2D CNN). In short sMAE pre-training improves phenotyping performance against all models, notably the two most performing gDL models in [Fawaz et al 2021], Spherical Unet and MoNet, by a large margin (apart for MoNet on PMA 0.63 MAE against 0.64 MAE, but with larger std).
>
> *Results and data split*: We apologise for overlooking the reporting of standard deviations of (v)sMAE reconstructions in Tables 3 and 4. They are now included in the tables. Given the limited time available for the rebuttal, we did not manage to run the statistical significance performance test for the many experiments reported in our submission. However, we are still working on it and will update the paper accordingly. All the phenotyping prediction experiments on the dHCP dataset, including the surface CNNs in Table 5, use the same train/val/test split as introduced in [Fawaz et al]. For the HCP dataset, the same train/val/test split is used between the different data releases (3T, 7T). We made this point clearer in Appendix A.1.
>
> *Data efficiency*: In Figure, we report the training loss and validation errors of different SiT models: trained from scratch, following MPP pre-training and following sMAE pretraining. The training logs demonstrate the fast convergence of the finetuned models (either from MPP or sMAE) for the difficult task of GA prediction. The data efficiency was also demonstrated through the transfer learning experiment in Figure 3.b, where a sMAE pre-trained SiT model on a large dataset (e.g UKB) is less data hungry for finetuning on phenotyping prediction tasks. Indeed, with 50\% of the data, the SiT model achieved $\leq 1$ MAE for scan age prediction, i.e. a $\sim 30\%$ improvement compared to a SiT trained from scratch, with the same amount of data. We believe that this result is particularly interesting in medical settings, where datasets are usually small in size.
>
> Additionally, we added a section to discuss memory requirements and computational cost of the method in Appendix A.2. However, there is an obvious additional cost of pre-training deep learning models. For instance here, sMAE pretraining on 4 cortical metrics typically takes 2 hours on a standard GPU hardware with a 128 batch size. Training vsMAE takes substantially more time (around a day or 50k iterations with a batch size of 128) because the cortical signal on task fMRI data is noisier to model. However, we believe that providing the checkpoints for these models would benefit the community and the required time to finetune new models, especially on smaller datasets.

---

> ### Author Response · Authors · 2024-03-18
>
> **Other comments:**
>
> *Concatenation across vertices*: We apologise for the lack of clarity in the description of the concatenation process. In short, once the triangular patches of imaging features (e.g. myelin, curvature, sulcal depth, and cortical thickness maps) are extracted from the input mesh (Figure 1.A.2), the vertices that fall into the same patch are concatenated along the feature dimensions. In practice, this means that the feature values corresponding to the vertices of a given patch are concatenated across channels. This creates a patch representation of size $45 \times C$, where C is the number of channels and $45$ corresponds to the number of vertices in a patch with ico3 sampling. Then this patch representation is projected to a token embedding of dimension $D$ with a linear layer. There is no need for a particular ordering of the imaging features/channels, however, the order of imaging features/channels must be maintained such that the concatenation process is consistent during training. More details are reported in [Dahan et al 2022, MIDL]
>
> *Reconstruction error*: We quantified the reconstruction quality for the (v)sMAE models for different masking ratios and reported the reconstruction loss in Table 3 and Table 4.

---

### Official Review · Reviewer_zph4 · 2024-02-28

**Confidence:** 3
**Preliminary Rating:** 4
**Final Rating:** 5

**Summary:**

This paper proposes two novel methodologies for multivariate and spatio-temporal pre-training of cortical signals over regular icosahedral grids: (1) surface Masked AutoEncoder (sMAE) and (2) video surface Masked AutoEncoder (vsMAE). The authors are building upon their previous work, SiT. The key idea in this paper is learning robust and generalisable representations through masking and reconstructing the surface data.

**Strengths:**

Very well written article with very clear context and motivation.

While whole brain is a more holistic approach to study the brain, extracting the information from the cortical folding is still one of the most important problems in brain image analysis and any approach improving cortical data analyses is valuable.

The proposed methods are building upon recent, SOTA spatio-temporal approaches such as VideoMAE, masked patch prediction.

The study uses publicly available datasets (HCP, UK Biobank), code and pre-trained models may be available upon publication (placeholder in paper).

**Weaknesses:**

The paper lacks comparisons of baseline models, focusing solely on ablation studies. Incorporating even a basic comparison, such as phenotype predictions using a group level cortical surface atlas (Schaefer et al.), could significantly enhance the understanding of the value added by this and other attention based reconstruction methods. Adding some discussion on this point could be helpful.

**Detailed Comments:**

It would be helpful if the authors discuss the computational cost of the proposed approach.

**Justification Of Final Rating:**

The authors have thoroughly addressed each of my comments and made appropriate updates to the relevant sections of the paper. With my concerns addressed, I am pleased to increase my rating, moving towards strong acceptance.

**Justification Of The Preliminary Rating:**

The paper addresses a topic of significant interest within the research community, indicating its relevance and potential impact on the field. The innovative methodologies proposed for the pre-training of cortical signals underscore the novelty. Extensive experiments for various parameters strengthens the study. Furthermore, the authors' track record and their foundation in previous work, such as SiT, suggest they are at the forefront of this area of study.

**Questions To Address In The Rebuttal:**

In Section 4. Experimental Methods, the sentence 'investigating whether the encodings were neurobiologically meaningful' raises a question: Does achieving improved performance necessarily indicate that the encodings are biologically plausible?

How does the phenotypic predictions from other representation learrning methods compare to (v)sMAE? For example, (not using icosahedrons, still surface data) Jung-Hoon Kim et al., Representation Learning of Resting State fMRI with Variational Autoencoder, Neuroimage (2021).

Minor:
Citation available for this claim, "In the present study, ico3 is used as it was found to be a good balance between patch resolution and computational requirements."?

---

> ### Author Response · Authors · 2024-03-18
>
> We would like to thank the reviewer for the time they took to review our submission and their very constructive comments. We are glad that the reviewer found the present problem important and our contributions valuable. We noted the important questions pointed by the reviewer and tried to address them in the following.
>
> **Points to address:**
>
> *Meaningful neurobiological encodings*:  The objective of the phenotype regression tasks was to validate whether our networks were – as a minimum - encoding phenotypically relevant information. Together with the strong generalisation performance of the model reconstructions, we would argue that this suggests that the model is learning the spatio-temporal dynamics of the brain. However, we completely agree that regression performance could be confounded by residual physiological noise signals, and as pointed out by the reviewer improved performance on phenotype prediction tasks does not necessarily equate to the biological plausibility of the learned encodings. We appreciate the opportunity to clarify our stance on this important aspect of our research. We plan to edit this statement to instead say ‘we investigated whether the encodings learned by our (v)sMAE models capture information relevant to phenotypic predictions and therefore may suggest an alignment with neurobiological patterns’. Investigation of how generalisable and interpretable the model encodings are (in the context of modelling the functional task) is out of the scope of this work (and the basis of a follow-up).
>
> *Phenotypic predictions and comparison with baseline models and other approaches*: We acknowledge the reviewer’s remarks on the lack of comparisons to baseline models for phenotyping prediction. In previous work (Fawaz et al 2022) we benchmarked the performance of surface CNNs (spectral convolution, spatial convolution) against classic ML driven by regional features derived from an atlas for the prediction of scan age (PMA) and birth age (GA) in the dHCP dataset. This showed that surface deep learning frameworks outperformed classic ML, especially for data that was not aligned (native configuration). Subsequently, in (Dahan et al 2022 MIDL 2022) we showed that the SiT improved the overall performance against these surface CNNs. In this submission, we have shown that the sMAE self-supervision of the SIT further improves performance - while, importantly, the same data splits were used throughout. We agree that a full appraisal of all these results would help readers of this paper and therefore have added these results to the appendix section in Table 5, with six baseline networks (five surface CNNs and one 2D CNN). In short sMAE pre-training improves phenotyping performance against all models, notably the two most performing gDL models in [Fawaz et al 2021], Spherical Unet and MoNet, by a large margin (apart for MoNet on PMA 0.63 MAE against 0.64 MAE, but with larger std).
>
> Of particular interest, we also compared our results to the performance of Projected 2D ResNet, where icospheric surface data were projected directly to a 2D image (similarly to the data representation in Jung-Hoon Kim et al. pointed out by the reviewer). Again, the finetuning of SiT models after sMAE pre-training outperforms the 2D CNN by a large margin on the two phenotyping prediction tasks (0.97/1.93 MAE vs 0.64/1.22 MAE or PMA/GA).
>
> Regarding the performance of phenotyping prediction of the vsMAE model on task fMRI data, we could compare the 0.39 Pearson correlation obtained here, to the 0.34 Pearson correlation obtained in [Improving Phenotype Prediction using Long-Range Spatio-Temporal Dynamics of Functional Connectivity, Dahan et al 2021] with a spatio-temporal graph convolution neural network model. In [Dahan et al 2021] functional connectome and associated time series were obtained from resting-state data from the HCP dataset following Group-ICA at different node resolutions and dual-regression of associated time courses.

---

> > ### Author Response · Authors · 2024-03-18
> >
> > **Other comments**:
> >
> > We apologise for the lack of clarity in the claim ‘In the present study, ico3 is used as it was found to be a good balance between patch resolution and computational requirements.’. We solely meant that ico3 patching allows patching the cortical surface at higher resolution (1280 patches against 320 as in [Dahan et al 2022, MIDL]), which allows visualising fine-grained patterns of spatio-temporal reconstruction activity, while still being not too complex to process with standard hardware. We reviewed this section to improve readability (appendix A.2)
> >
> > *Computational cost:* One of the advantages of the MAE framework lies in its light computational cost, as the encoder only processes the sequence of mask patches and the decoder is made shallower. We added a section to discuss memory requirements and computational cost of the method in Appendix A.2. However, there is an obvious additional cost of pre-training deep learning models. For instance here, sMAE pretraining on 4 cortical metrics typically takes 2 hours on a standard GPU hardware with a 128 batch size. Training vsMAE takes substantially more time (around a day or 50k iterations with a batch size of 128) because the cortical signal on task fMRI data is noisier to model. However, we believe that providing the checkpoints for these models, would benefit the community and the required time to finetune new models, especially on smaller datasets.

---

### Official Review · Reviewer_HzRw · 2024-03-01

**Confidence:** 4
**Preliminary Rating:** 5
**Recommendation:** Oral
**Final Rating:** 5

**Summary:**

This study introduces the surface Masked AutoEncoder (sMAE) and video surface Masked AutoEncoder (vsMAE) to address the challenge of encoding the spatio-temporal dynamics of brain activity, hindered by individual variations in the cerebral cortex.

**Strengths:**

Good model, good analysis, good idea.
I appreciated the use of UKB.
Very good work, solid.
********************************************************************************************************************************************************************************

**Weaknesses:**

Not sure what to suggest here.
********************************************************************************************************************************************************************************

**Detailed Comments:**

I suggest acceptance.

**Justification Of Final Rating:**

Nothing to add, I would like to stay on the Oral presentation.
********************************************************************************************************************************************

**Justification Of The Preliminary Rating:**

I liked the work, very solid.
Nothing to add.
********************************************************************************************************************************************************************************

**Questions To Address In The Rebuttal:**

none

---

> ### Author Response · Authors · 2024-03-17
>
> We would like to thank the reviewer for their careful review of our submission and their enthusiasm for our work and contributions.

---

### Meta-Review · Area_Chair_xbQc · 2024-04-04

**Recommendation:** Accept (Oral)
**Confidence:** 5

**Metareview:**

This paper proposes a surface masked auto encoder and video surface masked auto encoder for pretaining cortical signals over regular icosahedral grids.

Strengths:
+ Interesting novel extension of prior SiT work by the authors using SOTA pre-training strategies
+ New methods for cortical analysis of strong interest to community
+ Strong experiments using large public datasets, multiple tasks for evaluation
+ Well written paper

Weaknesses:
- Lacking statistical significance analysis
- The proposed pre-training approach comes with additional computational cost

After rebuttal, all reviewers are agreement that this is a very strong paper; I therefore also recommend accept.

---

### Decision · Program_Chairs · 2024-04-06

Accept (Oral)